

# 50 years of firn evolution on Grigoriev Ice Cap, Tien Shan, Kyrgyzstan

Horst Machguth[1], Anja Eichler[2,3], Margit Schwikowski[2,3,4], Sabina Brütsch[2], Enrico Mattea[1], Stanislav Kutuzov[5], Martin Heule[6], Ryskul Usubaliev[7], Sultan Belekov[8], Vladimir N. Mikhalenko[9], Martin Hoelzle[1], and Marlene Kronenberg[1, *]

[1]Department of Geosciences, University of Fribourg, Fribourg, Switzerland
[2]Laboratory of Environmental Chemistry, Paul Scherrer Institute, Villigen PSI, Switzerland
[3]Oeschger Centre for Climate Change Research, University of Bern, Bern, Switzerland
[4]Department of Chemistry, Biochemistry and Pharmaceutical Sciences, University of Bern, Bern, Switzerland
[5]Byrd Polar and Climate Research Center, Ohio State University, Columbus OH, USA
[6]Radioanalytics, Paul Scherrer Institute, Villigen PSI, Switzerland
[7]Central Asian Institute of Applied Geosciences (CAIAG), Bishkek, Kyrgyzstan
[8]KyrgyzHydromet, Bishkek, Kyrgyzstan
[9]Moscow, Russia
[*]now at Axpo Holding AG, Baden, Switzerland

**Correspondence:** Horst Machguth (horst.machguth@unifr.ch)

**Abstract.** Grigoriev Ice Cap, located in the Tien Shan mountains of Kyrgyzstan, has a rich history of firn and ice core drilling starting as early as 1962. Until now, the time series ended with a core drilled in 2007. Here we extend the exceptional record and describe an 18 m firn core, drilled in February 2018 on the summit of Grigoriev Ice Cap, at 4600 m a.s.l. The core has been analyzed for firn stratigraphy, major ions, black carbon, water stable isotope ratios and total $\beta$-activity. We find that

the core covers 46±3 years and overlaps by two to three decades with legacy cores. A good agreement is found in major ion concentrations for the overlapping period. Concentration in ions, susceptible to being washed out, is reduced since the early 2000s. This indicates the onset of meltwater runoff. Apart from runoff evidence, however, the firn appears remarkably unchanged. We find little change in net accumulation since the 1980s. Firn temperatures fluctuate, with 2018 temperatures being the highest on record ($\sim$-1.6 °C at $\sim$17 m depth). However, temperatures in 2023 are again similar to the early 2000s

at $\sim$-2.5 °C. We hypothesize (i) that firn temperatures are stabilized by the removal of latent heat through lateral meltwater runoff, and (ii) that mass loss by runoff is compensated by an increase in snowfall. While data from a nearby weather station support the latter hypothesis, thorough testing of both hypotheses will require surface mass balance and firn modelling.

## 1 Introduction

Grigoriev Ice Cap has a history as one of Central Asia's reference glaciers for climate reconstruction and firn temperature

observations. In 1962 Soviet glaciologists drilled a series of ice and firn cores on the ice cap (Dikikh, 1965). Towards the end of the Soviet Union, Mikhalenko (1989) initiated detailed studies of mass balance and firn structure on the summit of the ice cap. Two firn cores were drilled in 1990 (Thompson et al., 1993, 1997) which became the starting point of further drilling





campaigns (Arkhipov et al., 1997; Mikhalenko, 1997; Mikhalenko et al., 2005; Arkhipov et al., 2004; Kutuzov, 2005; Takeuchi et al., 2014, 2019). Recent research on the ice cap has broadened to topics such as glacier dynamics and ice volume estimates

(Nagornov et al., 2006; Fujita et al., 2011; Lavrentiev et al., 2018; Van Tricht and Huybrechts, 2022).

Early work on Grigoriev focused on understanding the mass balance of plateau glaciers (Dikikh, 1965; Safonov, 1983; Mikhalenko, 1989). The thermal regime of the glacier and refreezing of meltwater received particular attention. The 1990 shallow drilling aimed at investigating the potential of Grigoriev ice cap to retrieve longer climate records (Thompson et al., 1997). The collapse of the Soviet Union, however, made plans of deep coring largely obsolete. Instead, drilling on Grigoriev

continued with a focus on the firn layer, similar to the earliest work by Dikikh (1965). The data from the various cores show that ice and firn are warming (Mikhalenko, 1997), provide insight into net accumulation and indicate a rather dry climate (Mikhalenko, 1989; Thompson et al., 1997; Arkhipov et al., 2004). The data have also been used to study atmospheric pollution (Usubaliev, 2003) as well as to estimate the future evolution of the glacier (Fujita et al., 2011; Van Tricht and Huybrechts, 2022). The exception from shallow coring is an 87 m core drilled to bedrock in 2007. The core was analyzed to quantify glacier and

vegetation changes over several centuries to millennia (Takeuchi et al., 2014, 2019).

The Grigoriev time series of ice and firn cores appears to be unique for Central Asia, but is also remarkable on a global level. Currently, the published record ends with the 2007 core. In February 2018 we visited the ice cap, drilled an 18 m firn core and initiated continuous, automated observations of firn temperatures. The goal of the drilling was (i) to extend the exceptional time series and (ii) to quantify how the firn, as well as the glaciochemical records contained therein, are altered by current

atmospheric warming. Grigoriev is an interesting example to study such changes as firn temperatures, while still cold, have been measured relatively close to temperate conditions in 1990 and the early 2000s. Here we present the core analysis and the firn temperature record. We combine new and legacy measurements to quantify changes in net accumulation rates, stratigraphy, archived glaciochemical information and firn temperatures since the cores drilled in 1990. Observed changes could provide insight how other, currently still colder drill sites, might react to continued future warming.

## 40   2   Grigoriev Ice Cap

The Grigoriev ice cap (sometimes also transliterated to *Gregoriev*) is located in north-eastern Kyrgyzstan, at 41.975 °N / 77.913 °E in the Terskey-Alatoo range of the inner Tien Shan mountains (Fig. 1). The region is only seasonally inhabited by livestock farmers and herders during summer. The area is rather dry with annual precipitation of 291 mm (mean 1930-1997) and 350 mm (mean 1998-2009) as measured at the Tien-Shan / Kumtor weather station at 3614 / 3660 m a.s.l., ∼20 km east

of Grigoriev (Engel et al., 2012). The station was moved a few kilometers in 1997, names and elevations refer to until the year 1997 / since 1997. Precipitation peaks during summer, only a quarter of annual precipitation on Grigoriev falls during the months of October to April (Mikhalenko, 1989).

The ice cap forms a relatively flat glacier plateau of ∼8 km$^2$, reaching from ∼4100 to ∼4600 m a.s.l. The summit of the ice cap forms a flat area of roughly 400 m in diameter. At this site most legacy cores have been drilled. Further cores have been

collected near the summit or at locations south of the summit (Fig. 1).




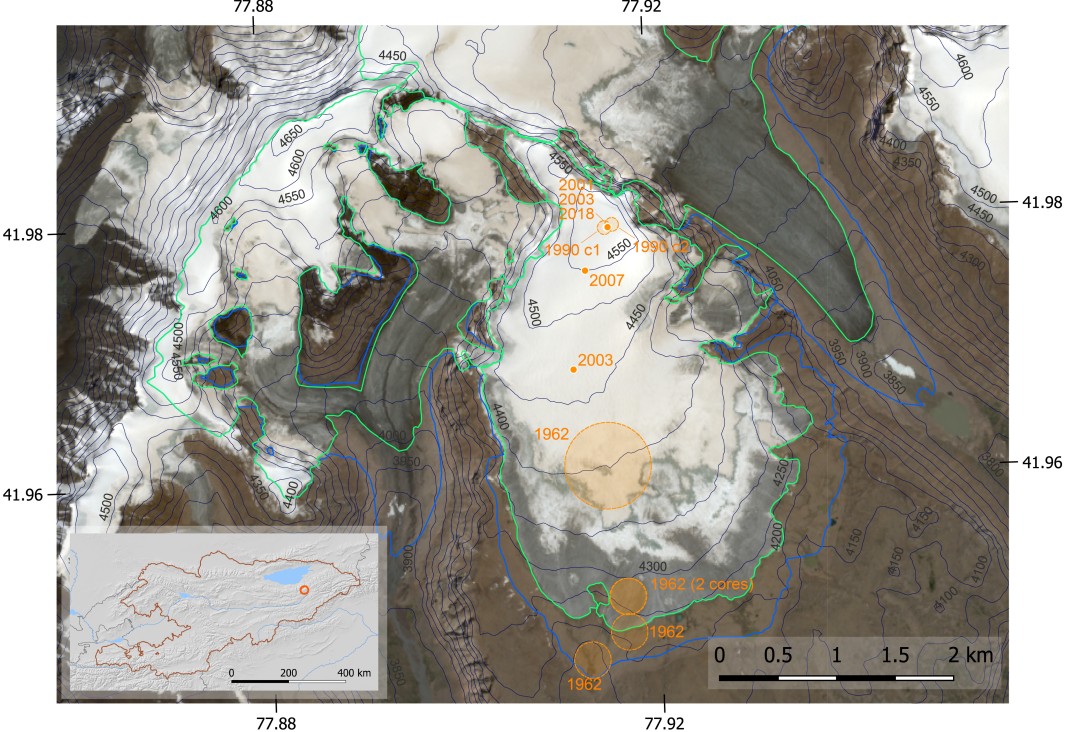

**Figure 1.** Grigoriev ice cap as seen on a Sentinel-2 L2A true color composite from 26 July 2021. The map shows the locations of the drill sites in orange (where exact coordinates measured shown as dots, approximate locations are shown with orange shading) and glacier extent for 1965 (blue) and 2022 (green). Elevation contours are based on SRTM digital elevation model (1 arc seconds horizontal resolution, vertical Datum EGM 96 geoid), coordinates are in decimal degree longitude/latitude. The inset map shows Kyrgyzstan with lakes in blue, the red circle denotes the location of Grigoriev ice cap.

Grigoriev ice cap is retreating and losing mass. Glacier retreat is most obvious at its southern margin. Since 1965 the ice margin has melted back by approximately 300 m or ~5 m yr$^{-1}$ (Fig. 1; Mikhalenko et al., 2005; Kutuzov, 2005; Kutuzov and Shahgedanova, 2009).

## 3 Data and Methods

### 3.1 Historical firn and ice core data

Starting in 1962, a total of seven core drilling campaigns provided data from Grigoriev ice cap (Table 1). The available data are diverse. Several cores were drilled on the summit, but various drill locations exist (Fig. 1). Cores reached to depths between 4 m and down to bedrock at almost 87 m. There is strong variability in the level of detail of core analysis. Density profiles and



firn temperatures are available most consistently. Not included in Table 1 is a coring expedition during the year 2002 which
suffered from challenging weather conditions and obtained little data.

**Table 1.** Overview of firn and ice core drilling campaigns that took place on Grigoriev ice cap since 1962. The column "Depth" also lists if a core was drilled mechanically "*m*" or thermally "*t*"; snow/firn pits are denoted by "*p*". Note that reported elevations of drill sites are of limited accuracy or might refer to different vertical datums. Over the time period 1990 to 2018, the summit plateau was likely always at roughly 4600 m asl. Originally reported elevations are listed while elevation and depth of cores drilled on the summit plateau are marked in **bold**.

| Year | Altitude [m asl] | Depth [m] and type of core | Density and stratigraphy | Major ions, water stable isotopes | Nuclear fallout | Temperature profile | References |
|---|---|---|---|---|---|---|---|
| 1962 | 4170, 4250, 4293, 4305, 4420 | $m$10, $m$10, $m$20, $m$10, $m$30 | - | - | - | Yes | Dikikh (1965) |
| 1987 | **4600** | $p$**4** | Yes, to 3.5 m | - | - | Yes, to 4 m | Mikhalenko (1989) |
| 1990 | **4660** | $m$**20**; $m$**16.43** | Yes, 20 m core | Yes | Yes, 20 m core | Yes | Thompson et al. (1993, 1997); Mikhalenko (1997); Arkhipov et al. (2004); Kutuzov (2005) |
| 2001 | **4625** | $m$**21.5** | Stratigraphy, no density | $\delta^{18}$O, $\delta$D; some ions (Table 2, Fig. 2; Usubaliev, 2003) | Yes, Fig. 6 in Mikhalenko et al. (2005) | Yes | Usubaliev (2003); Arkhipov et al. (2004); Mikhalenko et al. (2005) |
| 2003 | **4609**, 4440 | $m$**22.6**; $m$21.3; $t$**51**; $t$30 | Yes | $\delta^{18}$O, $\delta$D | Measured but data not found. | Yes, to 22 m and 45 m (summit), to 30 m at 4440 m asl (vicinity of 1962 location) | Mikhalenko et al. (2005); Kutuzov (2005) |
| 2007 | 4563 | $m$86.87 (bedrock) | Yes | $\delta^{18}$O, total particles; other ions unpublished | - | Yes | Shun et al. (2012); Takeuchi et al. (2014, 2019) |
| 2018 | **4600** | $m$**17.94** | Yes | Yes | Yes | Yes, constant logging. | Eichler et al. (2020), this study |

For the remainder of our study we mostly focus on the legacy summit cores drilled in the years 1990, 2001 and 2003. Stratigraphy has been reported for all of them, that is information about the presence of infiltration ice (i.e. ice lenses and



layers) and recrystallization ice (i.e. porous firn) is available at a vertical resolution of 1 cm or higher. Temperatures were measured in all the boreholes. Density records exist for all cores with the exception of one of the 1990 cores and the 2001 core
(Table 1). The existing density data for the other 1990 core are coarsely and irregularly spaced along the depth axis.

Stables isotopes $\delta^{18}$O and $\delta$D were measured in all cores. In the two 1990 cores NO$_3^-$, SO$_4^{2-}$, Cl$^-$ and impurities were analyzed, glaciochemical records exist also for the 2003 core (Usubaliev, 2003). No glaciochemical records are available for the other cores. Visible dust layers were recorded for most cores.

### 3.2 Drilling of the 2018 firn core

A firn core of 17.9 m in length was drilled on 7 and 8 February 2018 on the very top of Grigoriev ice cap (41.9791 °N / 77.9158 °E, ~4600 m a.s.l.). The core was of 9 cm in diameter and drilled using a Mark II Kovacs ice corer. The core was packed in polyethylene bags in segments of 0.16 to 0.57 m length (mean±1$\sigma$ = 0.34±0.09 m) and shipped in frozen condition for analysis to Paul Scherrer Institute (PSI) in Villigen, Switzerland.

### 3.3 Analysis of the 2018 firn core

The core segments were sampled in the coldroom of PSI at -20°C. First, each segment was photographed on a light table and the length, diameter, and stratigraphy (melt layers, dust layers) recorded. At a resolution of 1 cm, three classes were distinguished: snow, porous firn and ice. Where the content of a 1 cm interval was mixed, the percentages of the classes involved were estimated and noted down. Then, segments were cut using a modified band-saw setup (stainless steel saw blade; tabletop and saw guides covered with Teflon). Inner bars of the core segments with a cross section of ~1.9 x 1.9 cm, less prone to potential
contamination from drilling and handling procedures in the field, were cut and divided into ~5 cm samples. After measuring mass and dimensions of each sample for highly-resolved density determination, samples were placed into pre-cleaned 50 ml polypropylen tubes. For all core segments, two parallel samples were obtained. One sample was used for the analyses of water stable isotope ratios (WSI, $\delta^{18}$O and $\delta$D) and major ion (MI) concentrations and the other one for determination of the refractory black carbon (rBC) concentrations. A total of 2 x 361 samples was obtained for WSI/MI and rBC analyses.

Concentrations of MI (Na$^+$, NH$_4^+$, K$^+$, Mg$^{2+}$, Ca$^{2+}$, F$^-$, HCOO$^-$, CH$_3$COO$^-$, Cl$^-$, NO$_3^-$, SO$_4^{2-}$, (COO)$_2^{2-}$) were analyzed after melting the samples at room temperature using Ion Chromatography (IC; 850 Professional IC equipped with a 872 Extension Module Liquid Handling and a 858 Professional Sample Processor auto sampler, Metrohm) (Avak et al., 2019). The precision of the method is 5%. Determination of WSI ratios in the molten samples was performed using Wavelength-Scanned Cavity Ring-Down Spectroscopy (WS-CRDS, L2130-i Analyzer, Picarro) (Avak et al., 2019). The measurement uncertainty is
<0.1‰ for $\delta^{18}$O and <0.5‰ for $\delta$D.

We performed the analyses of rBC in the parallel samples, molten at room temperature and sonicated in an ultrasonic bath for 25 min, using a single particle soot photometer (SP2, Droplet Measurement Technology, Inc., Boulder) coupled to an APEX-Q jet nebulizer (Elemental Scientific Inc., USA) (Wendl et al., 2014). Precision of the method is ~10%.



### 3.4 Dating of the 2018 firn core and $\beta$-activity measurement

Precise dating of firn and ice cores is key to reconstruct past climate and atmospheric conditions. Age-depth scales of cores, covering recent decades, are often based on seasonally varying concentrations of impurities such as $NH_4^+$ (Eichler et al., 2000). If the seasonal signal is well preserved, annual layers are counted to establish an age-depth relationship.

For the Grigoriev core we applied annual layer counting using the seasonality in $NH_4^+$ concentrations and density records. The core is too short to reach the year 1963, usually identified by a tritium or $^{137}$Cs activity maximum from atomic bomb testing (e.g. Eichler et al., 2000). Instead we aimed to find the 1986 horizon, potentially marked by a maximum in total $\beta$-activity related to the fallout from the Chernobyl nuclear power station accident in April 1986.

For the total $\beta$-activity measurements, ten core samples in the depth range between 7 and 16 m and one blank sample (frozen ultrapure water) were prepared. Each sample consisted of 1 kg of ice, acidified in a 1 L plastic container with 0.33 ml of concentrated HCl (Hou et al., 2007; Eichler et al., 2020). Acidified samples were melted at room temperature and filtered using Macherey-Nagel MN616 LSA-50 cation exchange filters. Every sample was filtered three times and subsequently, the filters were dried at 60 °C, and analyzed for total $\beta$-activity. Each sample was measured six times 1 h at a Berthold LB 790 detector. Final $\beta$-activities were blank corrected (Eichler et al., 2020).

### 3.5 Firn temperatures and auxiliary temperature records

On 8 February 2018 we installed an automatic firn station on the summit plateau. The station's key component is a chain with 15 thermistors (type YSI 4460031), placed inside the borehole. At the time of installation the uppermost thermistor was located 0.4 m below the surface. Vertical spacing was 0.5 m until 5.4 m depth, followed by thermistors at depths of 7.4 m, 11.4 m, 17.4 m and 17.9 m. The chain was connected via a multiplexer to a data logger (CR-800, Campbell Scientific) attached to an aluminum pole at >2 m above ground. Furthermore, we installed a sensor measuring air temperature and a sonic ranger (SR-50, Campbell Scientific) to measure changes in surface height. The station was visited on 1 September 2018 and data were retrieved. It was noted that the multiplexer had stopped working on 28 March 2018, one and a half months after installation. The station was accidentally visited and photographed on 1 August 2021 (pers. communication, Lander van Tricht). In autumn 2022 the station was visited again and the multiplexer was replaced. On 1 August 2023 the station was once again visited, data were downloaded and the mast was extended. The station had been measuring continuously since the replacement of the multiplexer.

## 4   Results

### 4.1   Core stratigraphy, density and glaciochemical records

Figure 2 shows stratigraphy, density, measured concentrations of selected major ions, black carbon and $\delta^{18}$O. The core consists to 60.2 % of infiltration ice, the remainder being a ~65 cm snow layer at the surface and numerous layers of porous firn. The





density record indicates a mean density of $789\,\mathrm{kg\,m^{-3}}$. Given abundant infiltration ice, there is no obvious density gradient
with depth. However, layers of porous firn that are situated near the surface have lower densities than firn layers at depth.

Isolated samples show densities higher than the one of pure ice ($917\,\mathrm{kg\,m^{-3}}$; while the maximum measured in the core
is $996\,\mathrm{kg\,m^{-3}}$). These unlikely densities indicate uncertainties in the density measurements. We calculate the latter based
on estimated uncertainty of width and height (both $2 \pm 0.1$ cm), length ($5 \pm 0.1$ cm) of the small ice cubes whose weight
was determined with an uncertainty of 2% (all uncertainties expressed as $1\sigma$). Following the laws of error propagation, and
expressing overall uncertainty in percent, yields $7.6\,\%$ as $1\sigma$ uncertainty of individual density measurement.

We attribute fluctuations in major ion and black carbon concentrations to seasonal variations of atmospheric stability. Low
values occur in winter due to a decoupling of the high-altitude glacier site from the planetary boundary layer, where the sources
are located and which is thus more strongly polluted (e.g. Eichler et al., 2023). This behavior is the basis for dating by annual
layer counting (e.g. Eichler et al., 2000). $\delta^{18}O$ values also show fluctuations, but no clear seasonality. Observed maxima of these
species during the warm season are in accordance with observations of the seasonality of precipitation, which is predominant
in summer, and only a quarter occurs from October to April (Mikhalenko, 1989). In addition, surface melting and refreezing
of meltwater (relocation of the signal) might have caused a smearing of the signal.

Concentrations of the major ions $Cl^-$, $Na^+$, $SO_4^{2-}$, $Ca^{2+}$, $K^+$ and $Mg^{2+}$ are generally highly correlated among each other,
indicating joint transport and/or similar sources such as desert regions. The latter is in accordance with observations of atmo-
spheric dust entrainment occurring in central Asia primarily between spring and autumn (Grigholm et al., 2017). Remarkable
are the common depleted concentrations of major ions in the topmost 7 m (Fig. 2).

### 4.2 Dating of the 2018 core

Total $\beta$-activity background signals fluctuate between 18 and $60\,\mathrm{mBq\,kg^{-1}}$ (Fig. 3a). Two of the samples show above back-
ground activities of 83 and $108\,\mathrm{mBq\,kg^{-1}}$. The dust-normalized $\beta$-activity (ratio of $\beta$-activity/[$Ca^{2+}$]) also reveals a maximum
at the same depth (not shown). Retention of $^{137}$Cs, which is the main long-lived beta-emitter released by the Chernobyl acci-
dent, depends on the total quantity of insoluble matter. Since we did not measure the latter, we used $Ca^{2+}$ for normalization, as
signals in insoluble particle and $Ca^{2+}$ concentrations are typically well correlated (Bohleber et al., 2018). Based on measured
$\beta$-activities, we conclude that fallout from the Chernobyl accident is included in two samples. Thus, we assigned the depth of
12 m to the year 1986 (Fig. 3a).

Initial layer counting was based solely on seasonality of $NH_4^+$ concentrations and density and yielded a bottom age of 40
years with an estimated uncertainty of 10 years. Additionally using the Chernobyl 1986 reference horizon, we refined the
dating. Accordingly, the 17.8 m core covers the period from 1972($\pm3$) to 2018 and thus 46$\pm3$ years.

### 4.3 Combining the environmental record with earlier cores

We extended the environmental records by combining the records of the glaciochemical parameters analysed in both the 1990
and 2018 cores ($\delta^{18}O$, $Cl^-$, $NO_3^-$, $SO_4^{2-}$). Comparing the records where they overlap also indicates whether the dating of





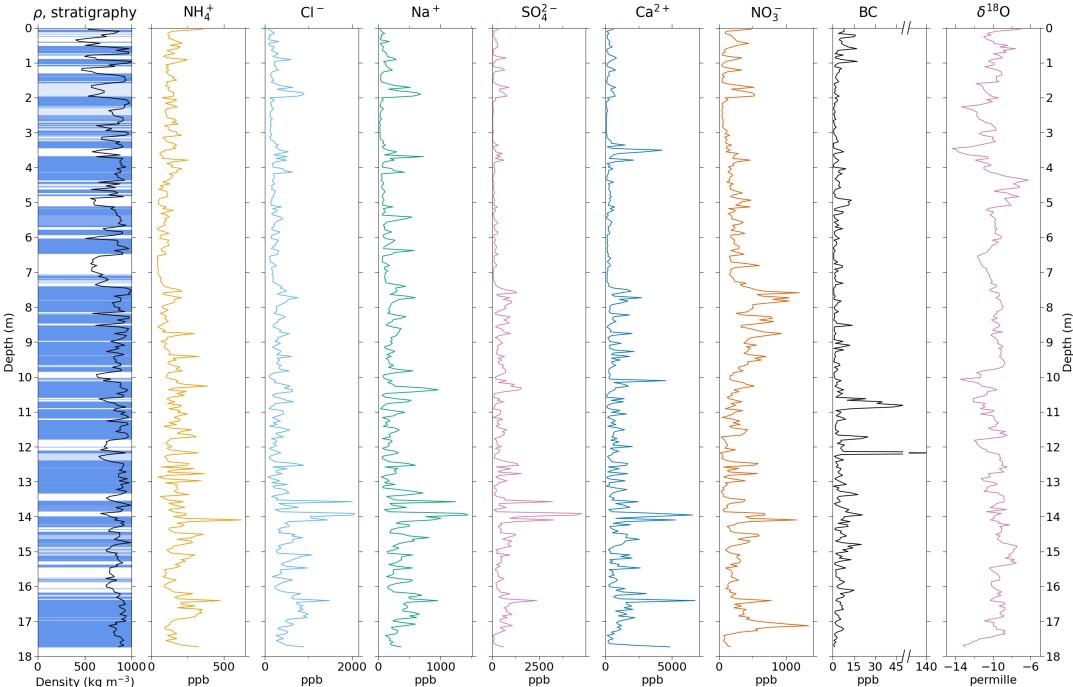

**Figure 2.** Firn stratigraphy, density, concentrations of selected major ions, black carbon and $\delta^{18}O$ in the 2018 Grigoriev core. Firn stratigraphy shows ice layers in blue and firn in white. Areas in light blue indicate regions of mixed firn and ice.

the 2018 core is compatible with the 1990 cores. For all four parameters, we find good agreement for the overlapping period (Fig. 4a to d).

### 4.4 Accumulation rate and melt proxies

We calculated mean net accumulation of 0.29 m w.e. yr$^{-1}$ for the 32-years period from 1986 (Chernobyl horizon) to 2018 (sur-
face) and of 0.31±0.02 m w.e. for the period from 1972±3 years (end of core) to 2018. We used annual layer dating and den-
sities (Fig. 3) to calculate annual and decadal net accumulation rates (Fig. 5a). Highest rates of net accumulation occurred be-
tween 1972 and 1979 (0.36 m w.e. yr$^{-1}$). Accumulation then dropped and reached a minimum in the 1990s (0.27 m w.e. yr$^{-1}$).
Afterwards, decadal mean net accumulation remained stable at ∼0.3 m w.e. yr$^{-1}$ for the years 2000 to 2018. As all other firn-
core based accumulation estimates for Grigoriev (e.g. Thompson et al., 1997; Mikhalenko et al., 2005), we do not consider
thinning of layers resulting from ice flow.

Figure 5b shows the proxy of annual Cl$^-$/Na$^+$ ratio together with the SO$_4^{2-}$ concentration. Layers strongly influenced by
melting are marked by Cl$^-$/Na$^+$ ratios exceeding the sea salt ratio of 1.78 and depleted SO$_4^{2-}$ concentrations. Since during
meltwater percolation a preferential elution of Na$^+$ compared to Cl$^-$ was observed (Eichler et al., 2001), a high Cl$^-$/Na$^+$ ratio





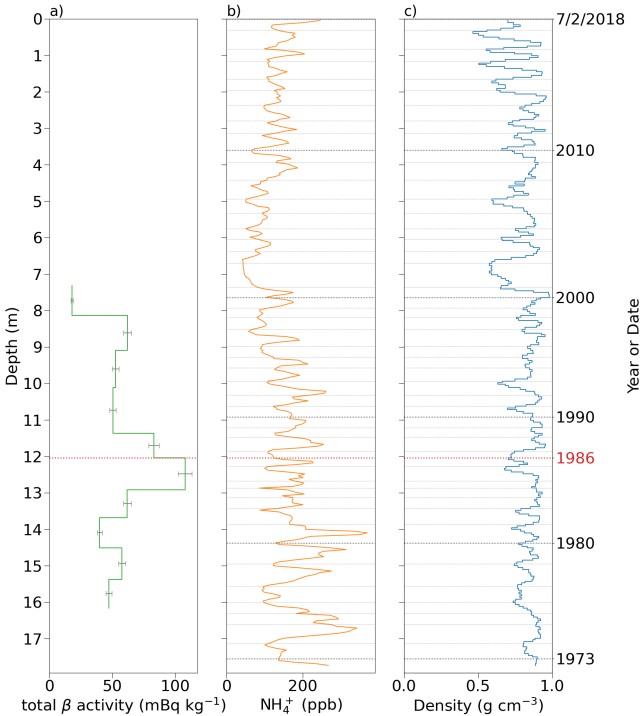

**Figure 3.** Dating of the 2018 Grigoriev firn core, showing depth and age axis. The 1986 reference horizon is highlighted in red. **a)** total $\beta$-activity with measurement uncertainties. **b)** Ammonium $NH_4^+$ concentration in ppb, 3-point moving average of the measurements in the core. **c)** 3-point moving average of density profile measured in the core. Thin horizontal lines in subplots b) and c) indicate annual layers, thick dotted lines show decadal boundaries.

is indicative for strong meltwater impact after ∼2005. The depleted $SO_4^{2-}$ since around the year 2001 suggests strong elution
of selected ions with meltwater.

## 4.5   Firn temperature (1990 - 2023) and firn structure (1990 - 2018)

Figure 6 shows firn temperatures on 28 March 2018 (last measurement before the failure of the multiplexer) and 1 August 2023
(most recent available measurement). The data are shown in the context of all legacy firn temperature measurements on the
summit. Also shown are the temperatures measured in the 2007 core that was drilled at 4563 m a.s.l., approx. 400 m south-east
of the summit.

Near-surface firn temperatures vary strongly because of measurements being taken at different times of the year. The 2018
data show substantial cooling of the near-surface during the winter months. Below ∼15 m depth the influence of seasonal
temperature fluctuations becomes absent (cf. Zagorodnov et al., 2006). For the remainder of this study, we focus on those
depth intervals as they allow determining long-term trends in firn temperatures. Figure 6 shows that the 2018 firn temperatures




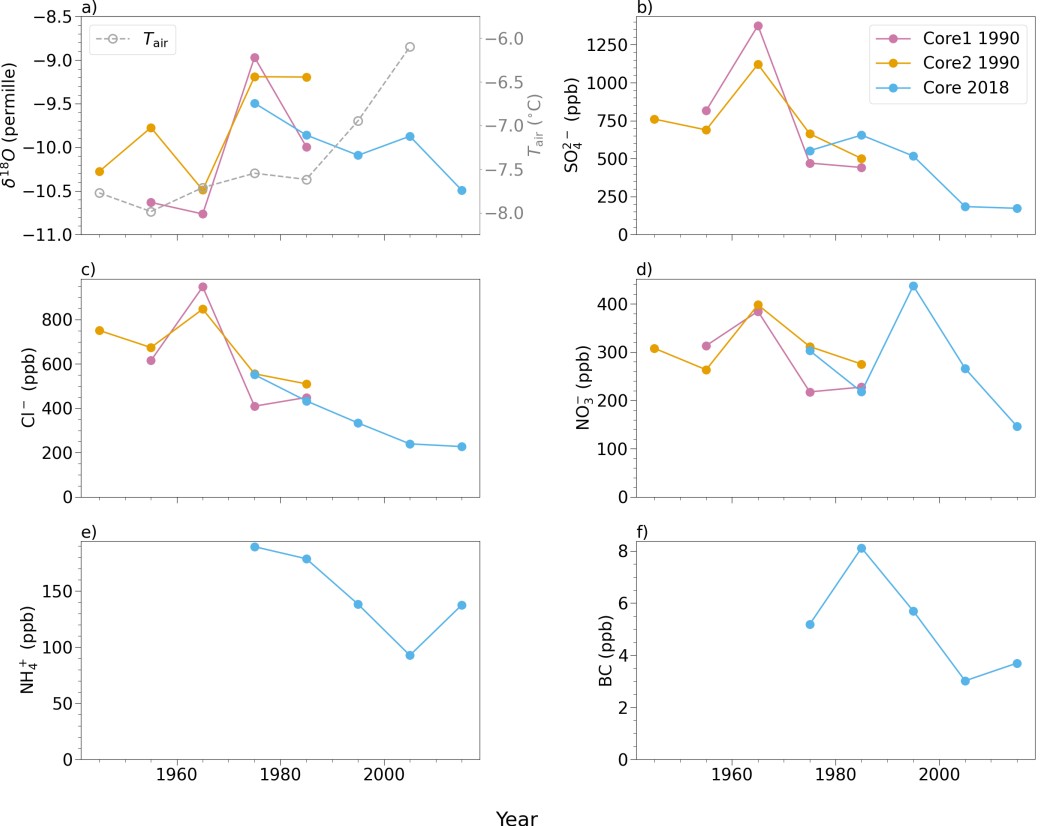

**Figure 4.** Comparison of decadal means in $\delta^{18}O$ stable isotope ratio, major ions and black carbon (BC), measured in the two 1990 cores as well as the 2018 core. Subplot **a** also shows decadal means of air temperature (1940s to 2000s) measured at the Tian-Shan / Kumtor weather station (data from Engel et al., 2012).

are the highest at -1.56 and -1.58 °C (17.4 and 17.9 m depth, respectively). In August 2023, however, we find firn temperatures of -2.53 °C and -2.46 °C (20.1 and 20.6 m depth, respectively). These firn temperatures fall within values typical for the early 2000s and indicate a cooling of ∼0.9 °C between March 2018 and August 2023. The change in depth is due to ∼2.7 m of snow accumulation between the two dates.

      We describe firn structure as content of infiltration ice per depth interval. Fig. 7 is designed after Kutuzov (2005) and
compares the 2018 content of infiltration ice to the legacy summit cores. The overall percentage of infiltration ice is similar for all cores. Only Core-1 1990 features a substantially higher percentage. The reasons for this are unclear as both 1990 cores were drilled in close vicinity. The difference could be related to Core 1 1990 having been analyzed *in situ* and Core 2 having been transported in frozen condition to cold room facilities in the US (Thompson et al., 1997). The 2018 core differs from the other cores mainly in the top ∼2 m where it has the highest ice content. This is also evident from mean densities of the near





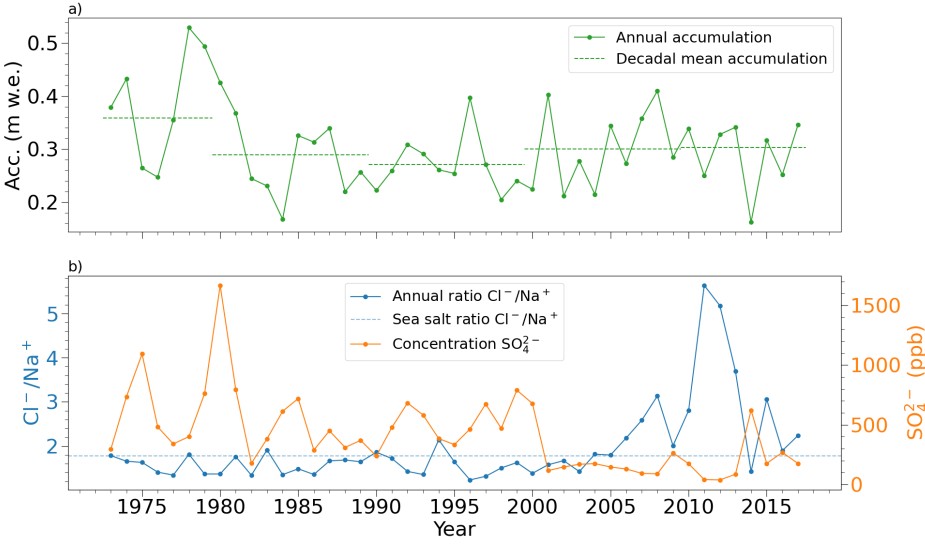

**Figure 5.** Accumulation rates and melt proxies. **a)** Annual and decadal rates of net accumulation. Note that the first (1972-1979) and last (2010-7/2/2018) "decade" do include fewer than 10 years. **b)** Melt proxies $Cl^-/Na^+$ ratio and $SO_4^{2-}$ concentration. For comparison, the sea salt ratio of $Cl^-/Na^+$ is shown.

surface layers. The 2018 core has a mean density of $709\,\mathrm{kg\,m^{-3}}$ and $774\,\mathrm{kg\,m^{-3}}$ for the top $2\,\mathrm{m}$ and $4\,\mathrm{m}$, respectively. The mean densities for the same depth intervals are 424 and $499\,\mathrm{kg\,m^{-3}}$ in the 2003 core.

## 5 Discussion

### 5.1 Accumulation

In 2018 and in the years before, the summit of Grigoriev ice cap was an active accumulation area. The same was the case from 2018 to 2023, as measured by the sonic ranger (not shown) which was the only device to work without interruption. Measured rates of net accumulation for the summit plateau (Sec. 4.4) are similar to previous estimates. Thompson et al. (1997) measured $0.33\,\mathrm{m\,w.e.\,yr^{-1}}$, for the time period 1962 to 1990. Arkhipov et al. (2004) and Mikhalenko et al. (2005) estimate net accumulation at $0.26\,\mathrm{m\,w.e.}$ (1991 to 2001) and $0.32\,\mathrm{m\,w.e.}$ (1963 to 1990). Takeuchi et al. (2014) measured $0.273\,\mathrm{m\,w.e.\,yr^{-1}}$ (mean 1967–2007) at their drill site, $\sim40\,\mathrm{m}$ lower than the summit. We also find temporal changes which are qualitatively similar to earlier studies. Arkhipov et al. (2004); Mikhalenko et al. (2005) state that net accumulation before 1990 was higher ($0.41$–$0.43\,\mathrm{m\,yr^{-1}}$) and then dropped to $0.35\,\mathrm{m\,yr^{-1}}$ for the time period 1990–2001. Our interpretation of the 2018 core indicates highest annual net accumulation in the 1970s, followed by a decrease throughout the 1980s and 1990s and a stabilization thereafter (Fig. 5a).

While runoff was considered absent towards the end of the 1980s (Mikhalenko, 1989), melt proxies, namely $Cl^-/Na^+$ ratio and $SO_4^{2-}$ concentration, indicate that melt has intensified and led to partial meltwater runoff starting sometimes between the





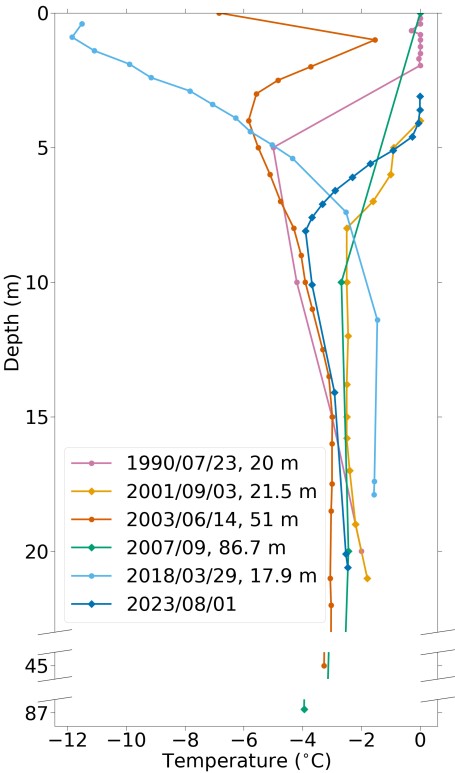

**Figure 6.** Firn temperatures measured on or near the summit of Grigoriev ice cap, 1990 to 2018. Date and depth of the borehole are indicated for each measurement. The 2018 borehole is represented with one measurement for 2018 and one for 2023. Note that the Y-axis is broken twice to show bottom temperatures of all temperature records.

years 2000 and 2005. As annual net accumulation appears unchanged since the early 1980s, the loss in meltwater needs to be compensated. Increased refreezing could compensate for runoff. However, stratigraphy and firn temperatures are mostly unchanged, with the exception of the uppermost two to four meters of the 2018 core that show higher ice content and densities than earlier cores.

We assume that compensating processes on the summit of Grigoriev result in mostly unchanged stratigraphy and net accumulation. Firstly, we hypothesize that increasing snowfall counterweights intensified melt and associated runoff. Indeed, Engel et al. (2012) report a change of mean annual precipitation at the Tien Shan / Kumtor weather station by +60 mm (+20%), comparing the two time periods 1930-1997 and 1998-2009. The authors state that the change is statistically significant. However, they caution that it is unclear whether the change is primarily caused by an increase in precipitation or by the station having being moved in 1997.

Secondly, we hypothesize that strong winter cooling is still relevant and contributes to a sustained refreezing potential and a largely unchanged firn stratigraphy. Mikhalenko (1989) noted that the shallow winter snow cover, together with cold winter air temperatures, leads to efficient winter-time cooling of the firn. This provides ample cold content to refreeze percolating





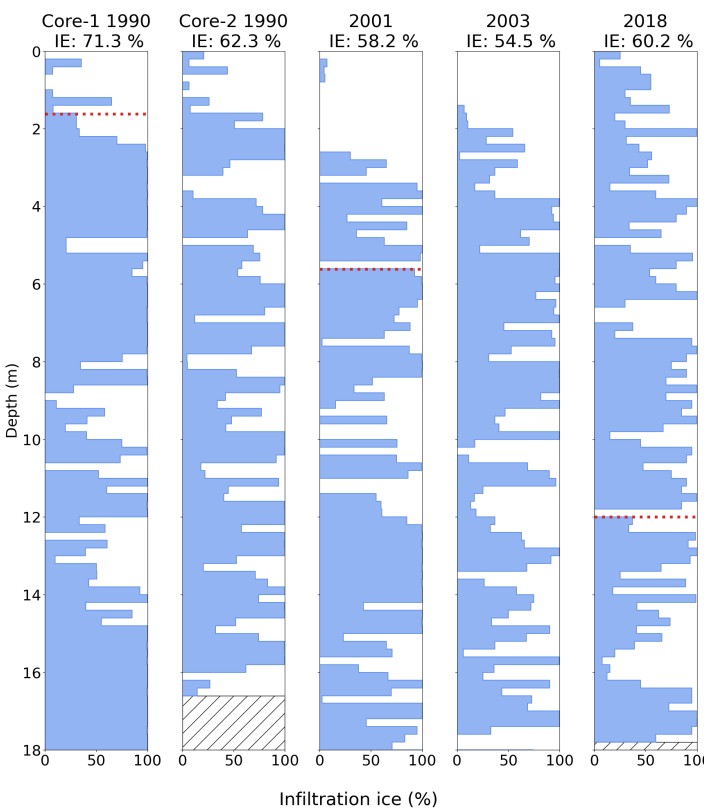

**Figure 7.** Infiltration ice (IE) in cores drilled on the summit of Grigoriev ice cap, 1990–2018. Shown is the percentage of infiltration ice per 20 cm depth intervals. Percentages in the titles indicate average content of infiltration ice for the top 18 m of each core (or total core length for shorter cores). The dotted red horizontal line indicates the depth of the 1986 Chernobyl horizon, where measured (Thompson et al., 1997; Kutuzov, 2005, and this study).

meltwater. Indeed, our own measurements show cold winter firn temperatures down to at least 5 m depth. However, no legacy
wintertime firn temperature measurements exist for Grigoriev, and we cannot assess to what degree winter firn temperatures have warmed.

To test these two hypotheses requires reliable data on the amount of snowfall in concert with a coupled surface energy balance and firn model. Existing modelling studies (Nagornov et al., 2006; Fujita et al., 2011; Van Tricht and Huybrechts, 2022) have not focused closely on Grigoriev firn properties and processes. A suitable approach was demonstrated by Kronenberg et al.
(2022), who modelled firn evolution of Abramov glacier, located in the Kyrgyz Pamir-Alai, approx. 600 km west south-west of Grigoriev. The motivation for their study was similar: a 2018 firn core and legacy data suggested a recent increase in net accumulation, seemingly contradicting increased melt and mass loss (Kronenberg et al., 2021).

The modeling study for Abramov glacier showed that increased melt was indeed masked by larger amounts of snowfall. Variations in refreezing played a relatively small role in the temperate accumulation area of Abramov. The study also showed



the limits of this mechanism and stated that net accumulation has finally begun to drop as snowfall cannot keep up any more
with ever increasing melt. This point might be close on Grigoriev as well, but our data indicate it has not yet been reached.

## 5.2 Firn temperatures

Kutuzov (2005) and Mikhalenko et al. (2005) compared Grigoriev firn temperature changes 1962 to 2003 at 4420 m a.s.l. They
show that the firn has warmed 1.5 and 0.9 °C at depths of 15 and 20 m, respectively. Arkhipov et al. (2004) analyzed firn
temperature changes on the summit until the year 2001. They conclude that the firn on the summit has warmed during the 11
years separating the 1990 and 2001 cores.

Temperatures measured in March 2018 are the warmest of all measurements. Temperatures in 2023, however, fall well within
earlier measurements. Unchanged firn temperatures stand in contrast to long-term observations from Alpine sites that show,
albeit starting from colder firn temperatures, a clear warming over comparable time frames (Vincent et al., 2007; Hoelzle et al.,
2011). We do not doubt the data from our thermistor chain. For example during the melt period of summer 2023, measured
firn temperatures in the top five meters deviate by <0.05 °C from 0 °C, which is the temperature one expects for firn that is in
contact with water. Why have firn temperatures reached a peak in 2018 and then cooled to earlier values? We assume that firn
temperatures on Grigoriev are on their way towards a fully temperate state at 0 °C, but get temporarily sidetracked by a variety
of mechanisms. The available data do not allow identifying the exact processes behind the observed fluctuations. Consequently,
three potential reasons are discussed in the following:

(i) The near-surface firn was at its densest and iciest in 2018. As a result, the permeability of the firn is reduced and meltwater
might discharge laterally rather than vertically. We have no direct evidence of surface runoff on the summit of Grigoriev, but
studies from other firn areas show that lateral runoff starts in the subsurface and becomes visible only after substantial distances
(e.g. several kilometers on the gently sloping surface of the Greenland Ice Sheet: Holmes, 1955; Clerx et al., 2022). Lateral
runoff would mean that less latent heat gets released at depth, inside the firn. If the icy firn gets efficiently cooled in winter, this
effect can lead to an overall cooling of the firn. Humphrey et al. (2012), for example, show that in the percolation zone of the
Greenland ice sheet, firn temperatures can decrease strongly with *decreasing* elevation. The reason is that vertical percolation,
and thus transport of latent heat, is more efficient at higher elevations where the firn is less icy and more porous. Also Arkhipov
et al. (2004) argues, on the example of Grigoriev Ice Cap, that one needs to be cautious when comparing temperatures measured
for firn of different ice content.

(ii) A number of firn temperature measurements (1990, 2001, 2023; Fig. 6) show a profile where below ~15 m depth the
firn gets warmer with depth. Such a temperature profile is unexpected as Takeuchi et al. (2014) measured a decrease of firn
temperatures all the way to the bed where the ice is at -3.95 °C (Fig. 6). The temperature profiles of 1990, 2001 and 2023 could
be explained by temporary and relatively deep firn warming, followed by a few years of cooling from the surface. A possible
mechanism could be very deep percolation and refreezing of meltwater during exceptionally warm summers. Crevasses could
serve as potential pathways for very deep percolation. Given the gentle and uniform warming over larger depth intervals, this
would need to happen at a certain distance from the drill sites. Any direct influx of water into or close to a borehole would





result in strong and sudden changes to the firn temperature profile (e.g. Machguth et al., 2016). However, this explanation is largely speculative as there is no data to show whether such events actually took place.

(iii) The nearby Tien Shan / Kumtor weather station shows strong increase in air temperature since its installation in 1930 (Engel et al., 2012). For the years 1998 to 2009, however, there appears to be no trend. While the latter could explain the lack in firn warming since around the year 2001, the data quality of the Tien Shan / Kumtor station is poorly known (see Secs. 2 and 5.1).

### 5.3   Suitability as environmental archive

**5.3.1**   $\delta^{18}$O

Based on the 1990 core, Thompson et al. (1997) proposed $\delta^{18}$O as $T$ proxy at the study site and interpreted the $\delta^{18}$O increase from 1940-1990 (Fig. 4a) as $T$ increase. The long-term $\delta^{18}$O record of the 2018 core does not show a trend, whereas air temperatures at the close-by Kumtor station increased. Engel et al. (2012) show a particularly pronounced change in air temperatures towards the end of the 1990s. While the same data issues apply as explained in Secs. 2 and 5.1, it is clear that air temperatures

at the weather station and $\delta^{18}$O on Grigoriev do not align (Fig 4a).

    The 1990 and 2018 $\delta^{18}$O data differ on average 0.3‰ (Fig. 4a), maybe due to slightly different locations of the two drilling sites. Differences between the two 1990 cores, drilled ∼50 m apart (Thompson et al., 1997), exceed this value for some decades. The absence of an increase in the $\delta^{18}$O values between 1970 and 2018 (measured in the 2018 core) could potentially be related to partial runoff of summer meltwater (revealing higher $\delta^{18}$O values) during the 2000s and 2010s or a relocation to deeper

layers and partial refreezing. Also a shift in seasonality cannot be excluded. Indeed, Engel et al. (2012) state that since 1998, a larger fraction of annual precipitation falls during summer (the aforementioned reservations towards data quality of the Tien Shan / Kumtor station apply also here).

### 5.3.2   Major ions and black carbon

Contrary to $\delta^{18}$O, seasonality in concentrations of most of the major ion is still visible, since they are not smoothed during firn

metamorphosis as $\delta^{18}$O is. In general we observe lower concentrations in the topmost 7 m, that is during the 2000s and 2010s. The reason could be meltwater-induced relocation and/or changes in the source strength. Increased air temperatures (Engel et al., 2012) likely have caused stronger melting, preferential elution, and possibly removal of ions by runoff. Consistent with the observation is that $NH_4^+$, $NO_3^-$ and BC concentrations are less depleted. Indeed, they are known to be less prone to removal by meltwater due to their location in the ice matrix or in the case of BC, to water insolubility (Moser et al., 2023). In addition,

emissions have changed as discussed below. Concentrations of dust-related highly correlated ions $Na^+$, $K^+$, $Mg^{2+}$, $Ca^{2+}$, $Cl^-$, $SO_4^{2-}$ agree well between the 1990 and 2018 cores for the overlapping period (Fig. 4b, c) and generally reveal a strong decrease after the 1960s. This is in agreement with declining trends of dust storms in Central Asia during that time established from stations in Xinjiang and the Karakum desert (Grigholm et al., 2017).



Concentrations of pollutants of mainly anthropogenic origin peak between the 1970s-1980s ($NH_4^+$ and BC) or the 1990s
($NO_3^-$) (Fig. 4d-f). Concurrent maxima of these species have been obtained from other Central Asian firn or ice cores, such
as Inilchek (Tien Shan, Grigholm et al., 2017) and Belukha (Siberian Altai, Olivier et al., 2003; Eichler et al., 2009) or from
an Elbrus core (Caucasus, Preunkert et al., 2019). For $SO_4^{2-}$, all other Central Asian cores reveal a maximum in the 1970s
reflecting the $SO_2$ emission history from different former Soviet Union countries (Olivier et al., 2003; Grigholm et al., 2017).
The strong $SO_4^{2-}$ decrease at Grigoriev from the 1960s on and the high correlation with dust proxies such as $Ca^{2+}$ suggests
that at this site natural dust sources for $SO_4^{2-}$ dominate compared to anthropogenic ones (see above).

## 6   Conclusions

We measured major ions, water stable isotopes and firn temperatures in an 18 m firn core, drilled in 2018 on the summit of
Grigoriev ice cap, at 4600 m a.s.l. in the inner Tien Shan mountains. Glaciochemical analysis revealed a core-bottom age of
1972±3 years. Subsequently we analysed our data in the context of legacy cores drilled at the same location. There is good
agreement in major ion concentrations where the 2018 record overlaps temporally with legacy cores. Fluctuations in major
ions agree with other studies carried out in Central Asia.

We find firn temperatures similar to the early 2000s, but also show that temperatures fluctuated substantially over the past
five years. Starting at around the year 2000, major ions are depleted in the core which we interpret as evidence of increased
melt and the onset of runoff. Around the year 2005 we observe the beginning of an increase in the $Cl^-/Na^+$ ratio, indicative
of strong meltwater impact. We find that the firn is more icy and more dense in the top two to four meters, compared to cores
drilled in the 2000s. Firn stratigraphy at greater depth is largely unchanged. We hypothesize that near-surface infiltration ice
might support lateral runoff. The latter might result in the removal of latent heat and thereby contributes to stabilizing firn
temperatures.

Given the onset of runoff, one might expect a decrease in net accumulation rates. However, net accumulation has remained
largely unchanged since the 1990s. We hypothesize that mass loss by runoff is compensated by an increase in snowfall,
resulting in largely unchanged net accumulation. While meteorological data from the nearby Tien Shan / Kumtor weather
station supports this hypothesis, the quality of these data is poorly known. We suggest using combined glacier surface mass
balance and firn models to explore (i) the possible influence of near-surface stratigraphy on lateral runoff and firn temperatures,
as well as (ii) a potential increase in precipitation on net accumulation.

*Data availability.* All data used in this manuscript are accessible at https://zenodo.org/doi/10.5281/zenodo.10082960. Previous versions of
stratigraphy and firn temperatures of the 2018 core are available at https://zenodo.org/records/7113282



*Author contributions.* MK and HM organized and led the 2018 drilling campaign to Grigoriev Ice Cap. AE led the measurements of major ions, water stable isotopes and black carbon with support by SB and MK. The measurements of $\beta$-activity were planned and carried out by MHeu. HM, AE, and MK performed data analyses. HM led the overall analysis and interpretation and wrote the manuscript with major contributions by AE and MS. The legacy data were collected by SK, VNM, MK, EM, RU and HM. EM and HM curated the data. EM led the autumn 2022 field campaign with support by MHoe; SB obtained the logger data in autumn 2023. Most authors commented on the manuscript.

*Competing interests.* The authors declare that the research was conducted in the absence of any commercial or financial relationships that could be construed as a potential conflict of interest.

*Acknowledgements.* We greatly acknowledge fieldwork assistance by Jonas Wicky, Philipp Schuppli, Ivan Lavrentiev, David Sciboz, Ruslan Kenzhebaev, Erlan Azisov and Martina Barandun as well as radionuclide measurements by Max Rüthi. We thank Lonnie Thompson and Ellen Mosley-Thompson for background information on the 1990 drill campaign and data. We thank Lander van Tricht for sharing photos of our firn station taken during his fieldwork in 2021. This study is funded by the Swiss National Science Foundation (Grant 200021_169453), the project CICADA (Cryospheric Climate Services for improved Adaptation; contract no. 81049674) and CROMO-ADAPT (Cryospheric Observation and Modelling for lmproved Adaptation in Central Asia; contract 81072443) between the Swiss Agency for Development and Cooperation and the University of Fribourg, as well as by the project GEF-UNDP-UNESCO (Strengthening the resilience of Central Asian countries by enabling regional cooperation to assess glacio-nival systems to develop integrated methods for sustainable development and adaptation to climate change; contract 4500484501) between UNESCO Almaty cluster office and Unversity of Fribourg. HM acknowledges support by the European Research Council (ERC) under the European Union's Horizon 2020 research and innovation programme (project acronym CASSANDRA, grant agreement No. 818994).



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
