# Peer review of "years of firn evolution on Grigoriev Ice Cap, Tien Shan, Kyrgyzstan"

_EGUsphere, 2023_

## Referee Comment (RC1)

Review of the manuscript: **50 years of firn evolution on Grigoriev Ice Cap, Tien Shan, Kyrgyzstan**

by H. Machguth et al.

**General comments**

The manuscript from Machguth et al. reports and extension of available ice core records at the Grigoriev Ice Cap, analysing a new core collected in 2018 for firn stratigraphy, major ions, black carbon, water stable isotope ratios and total β-activity. They find a good correspondence in the period overlapping former firn cores, and a reduction in the concentration of major ions since the early 2000s, which they relate to a recent increase in temperature and melt water percolation. The firn stratigraphy was found unchanged, with the exception of the shallow layers, and after an increase in 2018, its temperature in 2023 was found similar to the early 2000s.

Then the authors discuss the results, in particular the temperature stabilization and the largely unchanged net accumulation rate, in spite of increased percolation, formulating plausible hypotheses that might explain the observations.

In my opinion, the paper is well written, concise and clear. It requires only small formal adjustments and some integrations, as detailed in the specific comments. In particular, a better description and discussion is required for the measurement techniques and instruments, and some assumptions deserves further details. Formulated hypotheses are completely agreeable, however I think that the authors could include other hypotheses (e.g. for the stable net accumulation I would add the effect of snow metamorphism on wind drift) and possibly use their own data in support. For example, are the SR50 data useful for evaluating snow accumulation, redistribution and ablation?

Overall, my opinion is that this paper is publishable after a minor revision.

**Specific comments**

L7 - 'the firn appears remarkably unchanged': which features are unchanged?

L19 - I suggest adding 'thermal regime' as a topic of recent studies

L39 - insights 'into' how other…

L62-63: here the authors report that two categories are recognised, i.e. infiltration ice and recrystallization ice. However, the occurrence of surface melt and significant percolation suggest the likely occurrence of other type of ice formations, for example melt and refreeze crusts (formed at the surface). Please clarify.

L63 - what is the measurement technique for firn temperature? Was it homogeneous among compared cores? What was the measurement error? (i.e. are there possible discrepancies related to the measurement techniques or instruments?)

L165 - I think this is a crucial assumption, is there any evidence of negligible ice flow at this site (measured or modeled)?

L180-181 - possible malfunction of the thermistor string? What was the trend in temperature before the multiplexer failure? Any impact from multiplexer failure?

L195 - how was the pole sustaining the sensors installed? I mean, was there a support at the bottom of the pole, to prevent sinking in the snow/firn? This could have significant impact in automatic snow depth measurements. Are there snow pits measurements and/or snow depth soundings that confirm SR50 readings and support discussions on snow melt and percolation?

L200 - Arkhipov et al. (2004) 'and' Mikhalenko et al. (2005) state that….

Figure 7 - in the caption please add: Shown 'by the blue bars' is the percentage of infiltration ice….

L203 - in my opinion this discussion would benefit from a new figure (or a remake of figure 5a) that compares these different estimates of the accumulation rates, see also the following comment

L205 - in my opinion the authors should make a clear distinction between percolation and runoff. What do they mean with 'partial meltwater runoff'? Possibly, the authors means that percolated meltwater exceeded the irreducible water content and refrozen water in the firn layers at the top of the 2018 core? How can this be checked and/or quantified? Or is it only inferred by the melt proxies $Cl^-/Na^+$ ratio and $SO2^{-4}$ concentration shown in Figure 5b? A discussion is required of these aspects, because they are expected to affect accumulation rates, their historical trends, and possible alterations of former estimates at given depths/periods (I mean, is it possible that percolation and refreezing on firn layers of a given period lead to modification of former estimates of accumulation rates in that period?).

L206 - is it really a loss or a relocation? Please see the previous comment.

L232 - Another possible explanation is the positive correlation between air temperature and net accumulation at high-elevation sites, due to effects on surface snow metamorphism and lower susceptibility to wind erosion (please see e.g. Haeberli and Alean, 1985)

Haeberli, W. and Alean, J.: Temperature and accumulation of high altitude firn in the Alps, Ann. Glaciol., 6, 161–163, 1985.

L240 - were there possible alterations due to the drilling operations in 2018? Unfortunately, there were not 0°C temperature to be used for checking temperature measurements, such as in 2023.

L268 - in addition, the AWS data do not cover the period after 2009

L278 - it is actually a decrease if the authors refer to figure 4a

L279 - a runoff of meltwater from summer snow?

L280 - if a larger fraction of annual precipitation falls during summer, wouldn't the decadal means in δ18O be expected to increase?

L306 - here I would add a short sentence describing detected trends in major ions and their main causes.

L309 - please see the comment to L205. Runoff from the shallow firn layers?

L313 - contributes to stabilizing firn temperatures 'at the drilling site'

L315 - and/or reduced wind scouring?

---

## Referee Comment (RC2)

**Review for egusphere-2023-2722**

**Machguth et al - 50 years of firn evolution on Grigoriev Ice Cap, Tien Shan, Kyrgyzstan**

General comments:

The authors report results from a 18m firn core drilled at Grigoriev ice cap. This is especially interesting as it extends the present record of ice cores drilled on this ice cap. Evidence of increased melt is found in the top layers of the ice core when looking at major ions data; but firn temperature, after showing an expected increase in 2018, returns to early century values in 2023. The mechanism proposed to explain this dynamic is the presence of infiltration ice near the surface leading to lateral runoff instead of vertical percolation. The discussion also focuses on the net accumulation rate, which was found to be stable since the 1980s despite the aforementioned meltwater. The authors hypothesize the existence of a compensating mechanism that keeps the accumulation rate stable, such as the increase in precipitation.

The manuscript shows interesting data and the presented hypotheses are generally well supported. Overall it is well written, with some minor exceptions regarding a couple of paragraphs (see specific comments) which could be written in a clearer manner.

In my opinion, there is an issue that needs to be fixed regarding the weather station data: since this data is referenced and discussed many times in the text, it needs to be stated once and for all, and at the beginning of the manuscript, that this data must be taken with caution. It cannot be repeated in every paragraph like it is at present time in the discussion section.

The manuscript would also benefit from a more in-depth discussion regarding the total beta methodology and the major ion results.

Specific comments

L43-44. There is quite a difference in the data from before and after the moving of the weather station. I think this needs to be pointed out and further discussed since this data is frequently mentioned in the manuscript (see also general comments on this data). Engel et al (2012) point out that there is a significant difference in the two time series which should not be considered unitedly.

L61-68 "For the remainder of our study..." I think it would be best to rephrase here to make it more clear why the authors chose to focus on these specific cores. It is mentioned in the following lines, but in my opinion it is not clear that these are the reasons why.

L62. "that is information about..." wording is unclear: are saying that when you say stratigraphy has been reported you mean that for all cores you have information on the presence and position of infiltration and recrystallization ice, but not dust layers?

L82. misspelling of "polypropylene"

L102. From which part of the core did you prepare the samples for beta activity and which depth range did each sample cover? In my opinion, the explanation of this methodology could be slightly expanded.

L116. "the station was accidentally..." I don't understand the importance of reporting on this visit. I would also rephrase this paragraph to make it more immediate which are the time periods with available data.

L136. "only a quarter occurs from October to April" Would it be possible to sustain this claim but with more recent data from the weather station instead of citing a 30 year old paper?

L138. " generally highly correlated" I think this information needs to be better quantified.

Fig.3a I think it would be best to highlight the difference between measurements above and below the background for the total beta activity.

L145. "retention of 137Cs" This statement needs a reference for support.

L149. What is the error on the 12m depth which is arbitrarily assigned? I think it is important as this tie point is crucial in the dating of the ice core, which is thus affected by this error.

Fig4. Error needs to be shown for this data. It is difficult to compare the 3 cores without it. Additionally, Does it really make sense to show here the NH4 and BC data since there is no comparison with the other ice cores? It is stated that the 2003 core also had glaciochemical data, but none is shown here in comparison, why?

L156. " for all 4 parameters..." This statement needs to be better motivated.

L166-170. I think here it needs to be better explained why you are using Cl/Na ratio and showing SO4 concentrations. Just better wording to make it more clear.

L266. "for the years 1998..." This needs to be explained better since looking at Fig. 4a there is an apparent increase in air temperature.

L276. In my opinion, the discussion of the difference between 1990 and 2018 data would work better if moved to section 4.3

L276-282. This section is not very clear and could be phrased better to convey the message that dO18 is not conserved well probably due to percolating water.

L285. "In general we observe lower.." Is there a significant difference between concentration in the upper and lower part of the core? If so, how much? Are there also differences with respect to concentrations measured in other central Asian cores? I think this discussion would be interesting to add.

L286. "the reason could be..." In my opinion, the discussion would be more clear if the two hypotheses (meltwater-induced relocation and source strength change) were discussed more separately and a little more in-depth.

L288. "concentrations are less depleted" Less, but still depleted with respect to the lower part of the core?

---

## Author Comment (AC1)

**Reply to reviewers – Review 2**

*Dear editor, dear reviewers. We would like to thank you for your efforts regarding editing and reviewing of our study "50 years of firn evolution on Grigoriev Ice Cap, Tien Shan, Kyrgyzstan". Please find below in blue our replies to the reviewers' comments.*

The manuscript shows interesting data and the presented hypotheses are generally well supported. Overall it is well written, with some minor exceptions regarding a couple of paragraphs (see specific comments) which could be written in a clearer manner.

In my opinion, there is an issue that needs to be fixed regarding the weather station data: since this data is referenced and discussed many times in the text, it needs to be stated once and for all, and at the beginning of the manuscript, that this data must be taken with caution. It cannot be repeated in every paragraph like it is at present time in the discussion section.

The manuscript would also benefit from a more in-depth discussion regarding the total beta methodology and the major ion results.

We thank the reviewer for their positive evaluation of our work. We have addressed their remarks as described in detail below. We extended the description of the beta method: *"Every sample was filtered three times by filtering the filtrate twice. The filters were dried at 60 °C over night, and analyzed for total β-activity. Each sample was measured six times 1 h at a Berthold LB 790 detector and the results averaged. Final β-activities were blank corrected (Eichler et al., 2020) using the blank value of 33 mBq kg-1 of ultrapure water."* Furthermore, We have also merged the various statements on the data quality of the Tien Shan / Kumtor weather station, now appearing in Section 2: *"The Tien-Shan / Kumtor weather station is located at 3614 / 3660ma.s.l., ~20 km east of Grigoriev (Engel et al., 2012). The meteorological record starts in 1930, in 1997, the station was moved a few kilometers. The two names and two elevations refer to until the year 1997 / since 1997. The moving of the station introduced potential step-changes in meteorological parameters. For example, Engel et al. (2012) caution that it is unclear whether the above mentioned increase in average annual precipitation is primarily caused by a change in weather conditions or by the moving of the station. Because Tien-Shan / Kumtor is the only source of long-term meteorological measurements in the vicinity of Grigoriev, we use the data, qualitatively, to aid interpretation of the ice cores. We emphasize that any conclusions related to the station data need to be considered preliminary."*

L43-44. There is quite a difference in the data from before and after the moving of the weather station. I think this needs to be pointed out and further discussed since this data is frequently mentioned in the manuscript (see also general comments on this data). Engel et al (2012) point out that there is a significant difference in the two time series which should not be considered unitedly.

We agree with this remark and hope that the new paragraph on the two weather stations makes this clear (see our reply above).

L61-68 "For the remainder of our study…" I think it would be best to rephrase here to make it more clear why the authors chose to focus on these specific cores. It is mentioned in the following lines, but in my opinion it is not clear that these are the reasons why.

We now start the following sentence with *"We focus on these cores because ..."* followed by the explanations already there.

L62. "that is information about..." wording is unclear: are saying that when you say stratigraphy has been reported you mean that for all cores you have information on the presence and position of infiltration and recrystallization ice, but not dust layers?

*This has been clarified.*

L82. misspelling of "polypropylene"

*Corrected.*

L102. From which part of the core did you prepare the samples for beta activity and which depth range did each sample cover? In my opinion, the explanation of this methodology could be slightly expanded.

*In line 102 it is already written that the ten samples for beta activity cover the range 7-16 m. To clarify that the full depth range is covered by the ten samples, we added to L102: "…ten samples for beta activity covering continuously the range between 7 and 16 m depth and one blank sample…"*

L116. "the station was accidentally…" I don't understand the importance of reporting on this visit. I would also rephrase this paragraph to make it more immediate which are the time periods with available data.

*This remark had some importance in an earlier version of the manuscript but has since become irrelevant. We have removed the sentence. We have now added the information that after replacement of the multiplexer, the station recorded data again.*

L136. "only a quarter occurs from October to April" Would it be possible to sustain this claim but with more recent data from the weather station instead of citing a 30 year old paper?

*We have now complemented the citation of Mikhalenko (1989) with an estimate by Engel et al. (2012). Both numbers refer to the Tien Shan weather station.*

L138. " generally highly correlated" I think this information needs to be better quantified.

*We added the correlation coefficients for quantification: "Concentrations of the major ions $Cl^-$, $Na^+$, $SO_4^{2-}$, $Ca^{2+}$, $K^+$ and $Mg^{2+}$ are generally highly correlated among each other ($0.61<r<0.94$)…"*

Fig.3a I think it would be best to highlight the difference between measurements above and below the background for the total beta activity.

*The values in Fig. 3a represent blank-corrected beta activities. The blank value is 33 mBq $kg^{-1}$. The term "background activity" (18-60 mBq $kg^{-1}$) was just used to qualitatively describe the activities for all samples except the maximum values of 83 and 108 mBq-$kg^{-1}$. We agree with the referee that this was not fully clear from the present manuscript and adjusted this information in Lines 107, 143-144 and Fig. 3:*
*Line 107: "Final β-activities were blank corrected using the β-activity of ultrapure water of 33 mBq $kg^1$."*
*Line 143: "Blank corrected total β-activities vary between 18 and 108 mBq $kg^{-1}$ (Fig. 3a). Two of the samples show maximum activities of 83 and 108 mBq $kg^{-1}$."*
*Fig. 3: a) "… blank-corrected β-activity with analytical uncertainties…"*

L145. "retention of 137Cs" This statement needs a reference for support.

*We added the reference and adapted the sentence: "…The dust-normalized β-activity (ratio of β-activity/[$Ca^{2+}$]) also reveals a maximum at the same depth (not shown). Retention of β-activity on the filter, depends on the total quantity of insoluble matter (Picciotto et al., 1963)…"*

*Picciotto, E. and S. Wilgain. 1963. Fission products in Antarctic snow, a reference level for measuring accumulation. J. Geophys. Res., 68(21), 5965-5972.*

L149. What is the error on the 12m depth which is arbitrarily assigned? I think it is important as this tie point is crucial in the dating of the ice core, which is thus affected by this error.

There is no "error" on the 12 m depth.
The maximum of the beta activity is represented by two samples (11.3-12 m and 12-12.9 m depth, see Fig. 3a). Thus, we conclude that the year 1986 is included in both samples, and therefore we attributed the depth of 12 m to the year 1986. Lines 147-149 were changed accordingly:
*"Based on measured β-activities, we conclude that fallout from the Chernobyl accident is included in two samples (11.3-12 m and 12-12.9 m depth, see Fig. 3a). Thus, the depth of 12 m in between these two samples was assigned to the year 1986."*

Fig4. Error needs to be shown for this data. It is difficult to compare the 3 cores without it. Additionally, Does it really make sense to show here the NH4 and BC data since there is no comparison with the other ice cores? It is stated that the 2003 core also had glaciochemical data, but none is shown here in comparison, why?

We have updated figure 4 by (i) adding uncertainties where possible. (ii) we have changed the scale of $\delta^{18}O$ to allow direct comparison of *T* and $\delta^{18}O$ records. (iii) We prefer to leave $NH_4^+$ and BC data in the manuscript for discussing long-term trends in comparison to other Central Asian sites (L292-295). iv) The glaciochemical measurements carried out on the 2003 core offer few possibilities for direct comparison. *Usubaliev* (2003) focused more on metals, Ca and Na would be the only possibilities for direct comparison to our data, as well as $\delta^{18}O$ (*Kutuzov,* 2005). However, for a comparison like shown in Fig. 4, the 2003 core would need to be dated accurately. Initial alignment of the 2003 record against the 2001 core was been done by *Kutuzov* (2005) but more statistical work would be needed. We feel this is outside the scope of our study.

L156. " for all 4 parameters…" This statement needs to be better motivated.

The updated Figure 4 now shows that the parameters overlap within their bounds of uncertainty. We also mention this now in the text.

L166-170. I think here it needs to be better explained why you are using Cl/Na ratio and showing SO4 concentrations. Just better wording to make it more clear.

The deposition of sea salt (NaCl) is the dominant source of $Na^+$ and $Cl^-$ in the snow at the drilling site. Accordingly, firn core parts barely influenced by meltwater percolation show a $Cl^-/Na^+$ (mass) ratio of 1.78, corresponding to the sea-salt ratio. If firn layers are subject to meltwater percolation, $Na^+$ is preferentially removed with respect to $Cl^-$ (see e.g. Eichler et al., 2001). Then the $Cl^-/Na^+$ ratio exceeds the sea-salt ratio, as observed for many years after 2005 in Fig. 5b. Similar to $Na^+$, $SO_4^{2-}$ is strongly subjected to meltwater relocation as indicated by the low values in $SO_4^{2-}$ concentrations for years with maxima in the $Cl^-/Na^+$ ratio. We clarified the text accordingly.

L266. "for the years 1998…" This needs to be explained better since looking at Fig. 4a there is an apparent increase in air temperature.

Indeed, Fig. 7 in Engel et al (2012) suggests that, after a strong jump in air temperatures around 1998, there was no substantial change until the end of the record in 2009. The increase in decadal mean air temperatures shown in Fig. 4a agrees with this. Nevertheless, the data quality of the Tien Shan / Kumtor series is an issue. Since we were here trying to explain why firn temperatures in 2023 are similar to the early 2000s, it is even more of a problem that the available weather station record ends in 2009. For these reasons, we have removed the third potential explanation.

L276. In my opinion, the discussion of the difference between 1990 and 2018 data would work better if moved to section 4.3

We agree that there are a few pieces of quantitative information that could be placed in the results section. However, as most of the statements made here are discussion, we prefer to keep them where they are.

L276-282. This section is not very clear and could be phrased better to convey the message that dO18 is not conserved well probably due to percolating water.

We agree with the referee and in addition to deleting line 278-280 in response to referee 1, we add a sentence at the end of the paragraph: "*In conclusion, the strong increase of the air temperatures during the last decades is not reflected in the Grigoriev $\delta^{18}O$ record, implying that water stable isotopes are not controlled anymore by temperature variations at this site*".

L285. "In general we observe lower.." Is there a significant difference between concentration in the upper and lower part of the core? If so, how much? Are there also differences with respect to concentrations measured in other central Asian cores? I think this discussion would be interesting to add.

As suggested by the referee we investigated the differences between concentrations in the upper and lower parts of the core more quantitatively. Furthermore, the discussion of the two potential reasons for these differences was better separated and explained in more depth:

"*In general, we observe lower concentrations in the topmost 7 m, i.e. from the year 2001 on. Concentration ratios between the periods 2001-2018 and 1974-2000 are 0.31 ($SO_4^{2-}$), 0.36 ($Ca^{2+}$), 0.38 ($Mg^{2+}$), 0.46 (BC), 0.48 ($Na^+$), 0.51 ($K^+$), 0.58 ($Cl^-$), 0.6 ($NO_3^-$), 0.68 ($NH_4^+$), and 0.88 ($F^-$). Potential reasons for the decreasing concentrations are (1) meltwater-induced relocation and runoff and/or (2) changes in the emission source strength.*

(1) *Increased air temperatures (Engel et al., 2012) likely have caused stronger melting, preferential elution, and possibly removal of ions by runoff. This hypothesis is supported by the observation that $F^-$, $NH_4^+$, and $NO_3$- are less depleted. Those ions are known to be less prone to removal by meltwater due to their location in the ice matrix (Eichler et al., 2001; Moser et al., 2023).*

(2) *Concentrations of dust-related highly correlated ions $Na^+$, $K^+$, $Mg^{2+}$, $Ca^{2+}$, $Cl^-$, $SO_4^{2-}$ agree well between the 1990 and 2018 cores for the overlapping period (Fig. 4b, c) and generally reveal a strong decrease after the 1960s. This coincides with declining trends of dust storms in Central Asia during that time established from stations in Xinjiang and the Karakum desert (Grigholm et al., 2017). Concentrations of pollutants of mainly anthropogenic origin peak between the 1970s and 1980s ($NH_4^+$ and BC) or the 1990s ($NO_3^-$) (Fig. 4d-f). Concurrent maxima of these species have been obtained from other Central Asian firn or ice cores, such as Inilchek (Tien Shan, Grigholm et al., 2017) and Belukha (Siberian Altai, Olivier et al., 2003; Eichler et al., 2009) or from an Elbrus core (Caucasus, Preunkert et al., 2019). For $SO_4^{2-}$, all other Central Asian cores reveal a maximum in the 1970s reflecting the $SO_2$ emission history from different former Soviet Union countries (Olivier et al., 2003; Grigholm et al., 2017). The strong $SO_4^{2-}$ decrease at Grigoriev from the 1960s on and the high correlation with dust proxies such as $Ca^{2+}$ suggests that at this site natural dust sources for $SO_4^{2-}$ dominate compared to anthropogenic ones (see above).*

*In conclusion, despite the influence of melt, the general concentration trends of major ions and BC are consistent with observations and Central Asian ice core records, since emissions were highest during periods when melt influence was negligible. "*

L286. "the reason could be…" In my opinion, the discussion would be more clear if the two hypotheses (meltwater-induced relocation and source strength change) were discussed more separately and a little more in-depth.

See answer to comment above. The detailed discussion of the two hypothesis is now done more in depth and

L288. "concentrations are less depleted" Less, but still depleted with respect to the lower part of the core?

Yes, all are depleted to some degree. This has now been quantified while this entire part of the discussion was revised.

---

## Author Comment (AC2)

**Reply to reviewers – Review 1**

*Dear editor, dear reviewers. We would like to thank you for your efforts regarding editing and reviewing of our study "50 years of firn evolution on Grigoriev Ice Cap, Tien Shan, Kyrgyzstan". Please find below in blue our replies to the reviewers' comments.*

In my opinion, the paper is well written, concise and clear. It requires only small formal adjustments and some integrations, as detailed in the specific comments. In particular, a better description and discussion is required for the measurement techniques and instruments, and some assumptions deserves further details. Formulated hypotheses are completely agreeable, however I think that the authors could include other hypotheses (e.g. for the stable net accumulation I would add the effect of snow metamorphism on wind drift) and possibly use their own data in support. For example, are the SR50 data useful for evaluating snow accumulation, redistribution and ablation?

We thank the reviewer for their positive evaluation of our work. We have addressed their remarks as outlined below. We agree that susceptibility to snow erosion through wind depends on temperature and that the process needs mentioning. The effect on Grigoriev is difficult to quantify but so are our other hypotheses. We have added a brief discussion of the potential impact of snow erosion and temperature.

The SR50 data are not easy to use. Remember that air temperature measurements failed early, together with the thermistor chain. Hence, we cannot correct measured distances for air temperature fluctuations. Furthermore, in 2021 the distance between SR50 and snow cover fell below the minimum measuring distance, making it difficult to interpret the readings. For these reasons we would like to refrain from basing interpretation and conclusions on the SR50 data.

L7 - 'the firn appears remarkably unchanged': which features are unchanged? L19 - I suggest adding 'thermal regime' as a topic of recent studies

Has been changed to *"the firn's thermal regime appears remarkably unchanged."*

L39 - insights 'into' how other…

Changed

L62-63: here the authors report that two categories are recognised, i.e. infiltration ice and recrystallization ice. However, the occurrence of surface melt and significant percolation suggest the likely occurrence of other type of ice formations, for example melt and refreeze crusts (formed at the surface). Please clarify.

Agreed that other types of ice formation play a role. However, given that 55 to 70% of the cores are ice, with typical ice layers being several decimetres thick, more delicate features like melt and refreeze crusts or even wind crusts are of very limited relevance. We also have no records of very fine features from other cores. We have added *"Given the high ice content of all cores (see Section 4.5), already earlier studies focused on infiltration ice and less so on delicate features such as melt-freeze crust formed at former snow surfaces."*

L63 - what is the measurement technique for firn temperature? Was it homogeneous among compared cores? What was the measurement error? (i.e. are there possible discrepancies related to the measurement techniques or instruments?)

The technique of measurements was not homogeneous. We know that some of the early measurements were done by lowering a thermometer into the borehole, other measurements were done using thermistor chains. The main difference to earlier measurements is that we installed the thermistor chain permanently and measure continuously. Before we installed the thermistor chain, the longest measuring period was around a week, possibly also a few days longer. The different measuring techniques might impact the results. However, we have confidence in the earlier measurements for two reasons: (i) The number of past measurements seems high enough to conclude that our 2023 temperatures are within the range of earlier measurements. (ii) looking at our own data directly after installation, we see that about six to ten hours after installations, temperature at 17 m depth is already within 0.04 °C of the value around which it fully stabilized approximately two days after installation. Given that we know that earlier researchers were aware of the time it takes for temperatures to adapt, we are confident in the earlier measurements.

L165 - I think this is a crucial assumption, is there any evidence of negligible ice flow at this site (measured or modeled)?

Unfortunately, there is no measured ice flow at this site. *Van Tricht et al* (2022) modelled ice dynamics of Grigoriev ice cap. Their study indicates horizontal flow velocities close to zero at the summit. It is unclear how representative these simulations are, in particular over longer time periods and thus we feel that citing this study is not helpful in this context.

L180-181 - possible malfunction of the thermistor string? What was the trend in temperature before the multiplexer failure? Any impact from multiplexer failure?

There was no obvious trend before the failure, except for very close to the surface where the firn had warmed since installation. This agrees with our expectations as going from winter into spring, air temperatures had increased since the installation of the chain. We see no reason why the chain should have malfunctioned with the first multiplexer.

L195 - how was the pole sustaining the sensors installed? I mean, was there a support at the bottom of the pole, to prevent sinking in the snow/firn? This could have significant impact in automatic snow depth measurements. Are there snow pits measurements and/or snow depth soundings that confirm SR50 readings and support discussions on snow melt and percolation?

See Figure 1 below. The station had a rather large construction attached to the mast to minimize sinking into the snow and firn. This construction was likely not needed as the ice content of the firn is very high, even near the surface, and numerous thick ice layers prevent sinking in. It would be most challenging to dig into the very icy subsurface and no snowpits were dug. As there is virtually no chance for the station to sink in, we are confident in our measurements. We have slightly modified the text and made clear that our own accumulation measurements are not only based on the sonic ranger (who is too close to the surface since 2021) but also on manual measurements of snow height change.

[Figure]

*Fig 1: Grigoriev firn station on 8 February 2018, immediately after installation. Distance from snow surface to the horizontal boom is 2.74 m.*

L200 - Arkhipov et al. (2004) 'and' Mikhalenko et al. (2005) state that….

Corrected

Figure 7 - in the caption please add: Shown 'by the blue bars' is the percentage of infiltration ice….

Done

L203 - in my opinion this discussion would benefit from a new figure (or a remake of figure 5a) that compares these different estimates of the accumulation rates, see also the following comment

Unfortunately, we do not fully understand the comment. Does the reviewer mean that previous estimates should be compared with our estimates, which are shown in Fig. 5a? We would like to refrain from adding an additional figure or modifying Fig 5a. We believe that from the text it is clear enough that the various estimates are similar.

L205 - in my opinion the authors should make a clear distinction between percolation and runoff. What do they mean with 'partial meltwater runoff'? Possibly, the authors means that percolated meltwater exceeded the irreducible water content and refrozen water in the firn layers at the top of the 2018 core? How can this be checked and/or quantified? Or is it only inferred by the melt proxies $Cl-/Na+$ ratio and $SO2-4$ concentration shown in Figure 5b? A discussion is required of these aspects, because they are expected to affect accumulation rates, their historical trends, and possible alterations of former estimates at given depths/periods (I mean, is it possible that percolation and refreezing on firn layers of a given period lead to modification of former estimates of accumulation rates in that period?).

Previous studies showed that ions like $Na^+$, $Ca^{2+}$, and $SO_4^{2-}$ are enriched at the firn grain surfaces, from where they can be mobilized easily by small amounts of meltwater, and are preferentially eluted with meltwater (Eichler et al., 2001). The increased melt-proxy $Cl^-/Na^+$ and strongly depleted concentrations of $Ca^{2+}$, and $SO_4^{2-}$ suggest a significant influence of percolating meltwater on the chemical records after the beginning of the 2000s. Since no exceptional concentration peak is visible below the layers affected by meltwater, we assume that the meltwater containing the mobilized ions run off. Since this was most likely a small amount (other ions not disturbed), the effect on the accumulation rate is negligible.

We have expanded the discussion on these aspects, also on request from reviewer #2. We have also once again checked the entire manuscript to make sure that the usage of the terms percolation and runoff is unambiguous.

L206 - is it really a loss or a relocation? Please see the previous comment.

See our reply to the comment above.

L232 - Another possible explanation is the positive correlation between air temperature and net accumulation at high-elevation sites, due to effects on surface snow metamorphism and lower susceptibility to wind erosion (please see e.g. Haeberli and Alean, 1985)

Haeberli, W. and Alean, J.: Temperature and accumulation of high altitude firn in the Alps, Ann. Glaciol., 6, 161–163, 1985.

We agree to this important comment. While we cannot quantify any possible effect, we now mention the process and also refer to the study suggested by the reviewer. Towards the end of Section 5.1 we added: *"We note that warmer air and snow temperatures reduce snow erosion by wind (Haeberli and Alean, 1985). As wind certainly plays a role on the exposed summit of Grigoriev, it is possible that warmer air temperatures have reduced snow erosion. The possible effect is difficult to quantify and we emphasize that already in the past, accumulation took place mainly during the summer months."*

L240 - were there possible alterations due to the drilling operations in 2018? Unfortunately, there were not 0°C temperature to be used for checking temperature measurements, such as in 2023.

We do not fully understand what is meant with "positive alterations". Conditions during the drilling where very cold, below -20 °C. Until failure of the multiplexer we did not encounter 0 °C temperatures. The chain was produced in autumn 2017 and calibrated in a 0 °C water-ice bath prior to delivery. Maybe also our above reply on the observed adaptation time of firn temperatures help to clarify this point?

L268 - in addition, the AWS data do not cover the period after 2009

Correct, this should have been clearer. However, we have not added this information here as this comment is related to Reviewer's #2 request of more clearly communicating the limitations of the Tien Shan / Kumtor time series. We have added a paragraph focusing entirely on the Tien Shan / Kumto station (see answer to reviewer #2).

L278 - it is actually a decrease if the authors refer to figure 4a

This is correct and we have modified the text accodringly.

L279 - a runoff of meltwater from summer snow?

Yes. The sonic ranger measurements show how in the summers of 2018 and 2019 snow accumulates (increase of surface height) followed by surface lowering. We interpret the decrease in surface height to be partially due to melt (and partially compaction).

L280 - if a larger fraction of annual precipitation falls during summer, wouldn't the decadal means in δ18O be expected to increase?

Good point. The $\delta^{18}O$ record does not support the hypothesis of increased summer precipitation. We deleted this hypothesis in lines 278-280.

L306 - here I would add a short sentence describing detected trends in major ions and their main causes.

We agree with the referee and rewrote the first two paragraphs of the conclusion to include this information about major ions and also water stable isotopes:

*"We analysed major ions and oxygen stable isotopes in water in an 18 m firn core, drilled in 2018 on the summit of Grigoriev ice cap, at 4600 m a.s.l. in the inner Tien Shan mountains. Annual layer counting resulted in a core-bottom age of 1972±3 years. Subsequently we analysed our data in the context of legacy cores drilled at the same location. There is good agreement in major ion concentrations and $\delta^{18}O$ where the 2018 record overlaps temporally with legacy cores.*

*Firn temperatures measured in the borehole are similar to those from the early 2000s, but also show that temperatures fluctuated substantially over the past five years. During the last ~15 years the melt-index ($Cl^-/Na^+$ ratio) increased and major ion concentrations became depleted, which we interpret as evidence of accelerated melt and the onset of runoff. The general decrease of mineral dust related ions $Na^+$, $K^+$, $Mg^{2+}$, $Ca^{2+}$, $Cl^-$, and $SO_4^{2-}$, in the period 1972-2018 is in line with declining trends of dust storms in Central Asia. The concentration maxima of anthropogenically derived species $NH_4^+$, $NO_3^-$, and BC between the 1970s and 1990s are consistent with other studies carried out in Central Asia. Obviously, despite the influence of melt, the major ions and BC concentrations still reflect atmospheric emission trends, since emissions were highest during periods when melt influence was negligible. The strong increase of the air temperatures during the last decades is not reflected in the Grigoriev $\delta^{18}O$ record, implying that water stable isotopes are not controlled anymore by temperature variations at this site.*

*The firn was more icy and more dense in the top two to four meters, compared to cores drilled in the 2000s. Firn stratigraphy at greater depth is largely unchanged. We hypothesize that near-surface infiltration ice might support lateral runoff. The latter might result in the removal of latent heat and thereby contributes to stabilizing firn temperature…."*

To be consistent with the changes in the conclusions, the abstract was reworded as well:

*"…A good agreement is found in major ion concentrations and water stable isotope ratios for the overlapping period. Concentrations of black carbon and major ions, susceptible to being washed out, are reduced since the early 2000s. This indicates the onset of meltwater runoff. Nevertheless, general concentration trends of black carbon and major ions are consistent with observations and Central Asian ice core records, since emissions were highest during periods when melt influence was negligible. The record of water stable isotopes does not reflect the strong increase of air temperatures during the last decades, implying that water stable isotopes are not controlled by temperature variations at this site. Apart from runoff evidence, the net accumulation rates and firn temperatures appear remarkably unchanged since the 1980s. Firn temperatures fluctuate, with 2018 temperatures being the highest on record (~-1.6 ◦C at ~17m depth)…"*

L309 - please see the comment to L205. Runoff from the shallow firn layers?

Yes, please see our replies there. We have tried to make our interpretation clearer and added also here that we assume that runoff takes place laterally somewhere in the near-surface.

L313 - contributes to stabilizing firn temperatures 'at the drilling site' L315 - and/or reduced wind scouring?

Thank you for the important remarks on the effect of temperatures on wind. We have added *"and/or reduced wind scouring"* here.